# Increased Phenotype Severity Associated with Splice-Site Variants in a Hungarian Pediatric Neurofibromatosis 1 Cohort: A Retrospective Study

**DOI:** 10.3390/biomedicines13010146

**Published:** 2025-01-09

**Authors:** Klára Veres, Benedek Nagy, Zsófia Ember, Judit Bene, Kinga Hadzsiev, Márta Medvecz, László Szabó, Zsuzsanna Zsófia Szalai

**Affiliations:** 1Department of Pediatric Dermatology, Heim Pal National Pediatric Institute, 1089 Budapest, Hungary; szalai.zsuzsannadr@gmail.com; 2Faculty of Medicine, Semmelweis University, 1085 Budapest, Hungary; nagybenedek2004@gmail.com (B.N.); ember.zsofia@stud.semmelweis.hu (Z.E.); 3Department of Medical Genetics, Clinical Center, Medical School, University of Pécs, 7622 Pécs, Hungary; bene.judit@pte.hu (J.B.); hadzsiev.kinga@pte.hu (K.H.); 4Department of Dermatology, Venereology and Dermatooncology, Semmelweis University, 1085 Budapest, Hungary; medvecz.marta@semmelweis.hu; 5Department of Internal Medicine, Heim Pal National Pediatric Institute, 1089 Budapest, Hungary; szabo.laszlo.md@gmail.com; 6Department of Family Care Methodology, Institute of Health Science, Semmelweis University, 1085 Budapest, Hungary

**Keywords:** rare diseases, genotype–phenotype analyses, NGS, type 1 neurofibromatosis, splice-site mutations, *NF1* gene

## Abstract

**Background:** Neurofibromatosis type 1 (NF1) is a complex neurocutaneous disorder caused by pathogenic variants in the *NF1* gene. Although genotype–phenotype correlation studies are increasing, robust clinically relevant correlations have remained limited. **Methods:** We conducted a retrospective analysis of data obtained from a cohort of 204 Hungarian individuals, with a mean age of 16 years (age range: 1–33 years). The data were collected over 15 years. **Results:** Among the cohort of 204 patients, 148 subjects fulfilled ≥2 criteria established by the National Health Institute. Genetic testing was performed in 70 patients, with an 82.8% detection rate, of which 13 patients were excluded. Among the remaining 45 pathogenic variants, 17 (37.7%) frameshift, 11 (24.4%) nonsense, 8 (17.8%) splice-site, 4 (8.9%) missense mutations, and 5 (11.11%) copy number variations (CNVs) were detected. Café-au-lait macules were present in all patients (100%). Intracranial malformations were the second most common feature (55.6%), followed by Lisch nodules (35.6%), neurofibromas (33.3%), and skeletal abnormalities (31.1%). **Conclusions:** In our cohort, patients with splice-site variants (8/45, 17.8%) demonstrated a notably more severe phenotype compared to findings reported in other studies, with a high prevalence of plexiform neurofibromas (37.5%), intracranial findings (62.5%), skeletal abnormalities (50%), Lisch nodules (50%), and even pseudarthrosis (25%). Correlating with the literature, missense variants represented a mild phenotype, while patients with microdeletion syndrome revealed a more severe phenotype.

## 1. Introduction

Neurofibromatosis type 1 (NF1) (MIM #162200) is an autosomal dominant neurocutaneous disorder exhibiting a worldwide incidence of 1 in 2500–4000 individuals [1,2,3,4]. NF1 is caused by mutations in the tumor suppressor *neurofibromin* (*NF1*, NM_000267) gene located at 17q11.2 of chromosome 17. Due to the large size of the *NF1* gene, which encompasses 350 kb of the genome and contains 61 coding exons, it is susceptible to a wide range of mutations, leading to de novo mutations in up to 50% of cases. Approximately 85–90% of the mutations are point mutations, 5–10% are microdeletions, and 2% are exon deletions or duplications [1,2,5]. The disease typically progresses over time, starting in childhood, and shows nearly complete penetrance with advancing age [6,7,8]. According to the literature, 50% of patients demonstrate characteristic clinical signs by the age of 1 year, and 97% by the age of 8 years [1,3,8]. The literature suggests that NF1 patients have a reduced life expectancy of 15 years, compared to the general population, mostly due to vascular diseases and malignant tumors, including female breast cancer under 50 years and malignant peripheral nerve sheath tumors (MPNSTs) [2,9].

The criteria of the National Institutes of Health (NIH) for NF1 diagnosis established in 1987 were revised in 2021, to include six or more café-au-lait macules (CALMs, ≥5 mm in prepubertal individuals); axillary or inguinal freckling; two or more neurofibromas or one plexiform neurofibroma (PN); optic pathway glioma; Lisch nodules or choroidal abnormalities; a distinctive osseous lesion (e.g., sphenoid dysplasia, anterolateral bowing of the tibia, or pseudarthrosis of a long bone); and a heterozygous pathogenic *NF1* variant [8,9,10]. A diagnosis can be established if the patient meets at least two criteria. However, a child of a parent diagnosed with NF1 needs only one of the NIH criteria to merit a diagnosis of NF1 [9]. Naevus anaemicus, pseudo-atrophic macules, glomus tumor, scoliosis, and juvenile xanthogranulomas are also more frequently seen in NF1 patients [3,4,9,11,12,13,14].

*NF1* encodes neurofibromin, a multifunctional protein acting as a GTPase-activating protein (GAP) that negatively regulates the RAS/MAPK pathway activity by accelerating the hydrolysis of RAS-bound GTP [1,9,15]. Lately, MEK inhibitors that inhibit this pathway have been approved to be effective in the reduction in tumor bulk and for improvement in symptoms in cases of inoperable plexiform neurofibromas [1,16]. In April 2020, selumetinib, an oral selective MEK inhibitor was approved by the US Food and Drug Administration for the treatment of children with NF1-related symptomatic, progressive, inoperable plexiform neurofibroma, which can also be effective in the treatment of optic glioma [17,18]. Several other MEK inhibitors (binimetinib, mirdametinib, trametinib) and the tyrosine kinase inhibitor cabozantinib are also being investigated as medical therapies for NF1-related plexiform neurofibromas [1,18].

This ongoing exploration of targeted treatments has still reported only modest results, which highlights the importance of potential future therapies, such as gene therapy. This reinforces the significance of genotype–phenotype studies to tailor these interventions more effectively to individual patient profiles [19].

Due to the significant clinical and genetic heterogeneity of Neurofibromatosis Type 1, which establishes a clear genotype–phenotype correlation in NF1 patients remains challenging. Nevertheless, according to the growing popularity of genetic testing, more genotype–phenotype correlations have been elucidated.

The aim of the present study was to analyze the clinical features and genetic findings of patients with Neurofibromatosis Type 1 caused by mutations in the *NF1* gene, who were inpatients and/or outpatients in the Heim Pal National Pediatric Institute, Budapest, Hungary between 1 January 2010 and 30 October 2024.

We conducted a retrospective analysis of clinical, anamnestic, radiological, and genetic data of 204 individuals, who met at least one National Institutes of Health (NIH) criterion for NF1 diagnosis, including 148 subjects fulfilling at least two of the diagnostic criteria.

### Clinical Characteristics


**Skin manifestations**



**Café-au-lait macules (CALMs)**


Café-au-lait macules (CALMs) are the earliest manifestations of NF1, typically appearing in infancy [20,21,22,23,24]. By the age of one, 99% of NF1 patients exhibit six or more CALMs, which usually increase during early childhood before they stabilize or even fade [11,20,23,24]. While typical CALMs are ovoid with well-defined borders of 1–3 cm, and uniformly pigmented (slightly darker than the surrounding skin), the occurrence of significant variations in size, pigmentation, and shape is possible [11,20,21,22,25,26]. CALMs can occur almost all over the body but are not seen on the palms or soles of people with NF1 [11,25,26].


**Freckling**


Axillary and inguinal freckling (Crowe sign), characterized by small (1–4 mm), clustered, pigmented macules, is a pathognomonic feature of NF1 [11,20,24,25,26]. It typically appears between three and five years of age, often after the appearance of CALMs and before that of neurofibromas, and it presents in over 85% of NF1 patients by the age of seven [11,20,24]. Axillary or inguinal freckling is one of the seven cardinal diagnostic features of NF1 [25,26].


**Neurofibromas and Other Tumors**



**Neurofibromas**


Cutaneous neurofibromas (CNFs) are benign tumors of the dermis, frequently occurring in large numbers (hundreds or thousands) in NF1 patients [27,28,29]. The morphological presentation of CNFs is variable, encompassing nascent/latent, flat, sessile, globular, and pedunculated lesions. These lesions are typically soft in consistency, nontender on palpation, and range in size from 0.5 to 30 mm [20,28]. It is important to note that these tumors do not exhibit malignant potential [27,29]. Their prevalence increases from adolescence to adulthood [24,27,28], ultimately appearing in over 90% of affected adults [20].


**Plexiform neurofibromas**


Plexiform neurofibromas (pNFs), which affect from 20% to 50% of patients with NF1 [1,11,16,18,20,30,31], are histologically benign tumors of the peripheral nerve sheath, with the potential to transform into malignant peripheral nerve sheath tumors (MPNSTs) in approximately 8% to 13% of NF1 patients [32]. Plexiform neurofibromas are typically congenital, and grow slowly, except during periods of early childhood and pregnancy, when the tumor mass tends to increase more rapidly [1,20,30]. These tumors can lead to significant disfigurement, visible deformities, pain, and local compression of adjacent structures, resulting in functional impairment of nerves, vasculature, and airways [1,24,29,30]. Recent evidence indicates that individuals with non-mosaic large NF1 deletions exhibit a markedly higher tumor burden, which correlates with an increased propensity for malignant transformation into MPNSTs [24,32]. Magnetic resonance imaging (MRI) plays a critical role in the early detection and monitoring of asymptomatic internal pNFs, thus facilitating timely intervention and management.


**Ocular manifestations**



**Lisch nodules**


Lisch nodules, benign, well-defined, gelatinous pigmented lesions of the iris, are hallmarks of NF1 and its most common ocular manifestation [1,24]. Lisch nodules typically appear after the age of six, increasing in prevalence with age and reaching 90–95% by the age of 30 [33]. They are significantly more common (22.6% vs. 9.1%) in patients with frameshift mutations, as reported by Stella et al. [34], whose results are consistent with previous findings [35,36].


**Optic glioma**


Optic pathway gliomas (OPGs) are the most common type of glioma in NF1, affecting approximately 15% to 20% of young children with the condition, typically before the age of seven [20,24,37,38]. While the majority of patients are asymptomatic, only 30% to 50% develop symptoms, and of those, roughly one-third may require intervention [20,37]. OPGs are primarily low-grade pilocytic astrocytomas that can arise anywhere along the optic pathway, and lead to a range of clinical manifestations depending on the location of the tumor. Symptoms may include unilateral proptosis, decreased visual acuity, visual field defects, and, in rare cases, obstructive hydrocephalus with headache, nausea, and vomiting [37].


**Skeletal manifestations**


Skeletal malformations affect approximately 50% of NF1 patients and typically present in early childhood, including scoliosis, osteopenia, osteoporosis, tibial dysplasia, and pseudarthrosis (a false joint from long bone fracture nonunion) [20,24,38,39,40]. Scoliosis affects 21% to 49% of NF1 patients, often with rapid progression compared to the general population [38,39,41]. Long bone dysplasia is observed in 2% to 4% of patients, with congenital tibial dysplasia (CPT) more frequently involving the tibia [39,41]. Anterolateral tibial bowing and tibial dysplasia are present in about 5% of NF1 patients and are leading causes of pseudarthrosis [41]. Additionally, both children and adults with NF1 demonstrate decreased bone mineral content compared to age-matched controls, with osteoporosis developing earlier, especially in postmenopausal women [39].


**Neurological and brain MRI findings**


Neurological complications are among the most commonly observed findings in patients with NF1, affecting an average of 30% to 60% with learning and language difficulties, as well as symptoms of inattention, hyperactivity, or impulsivity [42,43]. Other notable neurological issues include epilepsy (often secondary to structural lesions), migraine-like headaches, brain gliomas, intracranial malformations, aneurysms, and moyamoya syndrome [42,44,45]. Magnetic resonance imaging (MRI) is the preferred modality for detecting alterations such as optic nerve gliomas and brain tumors [42,44]. A significant percentage of patients exhibit T2 hyperintensities in several brain areas, referred to as unidentified bright objects (UBOs), which remain of uncertain clinical significance and may resolve with age [42,44,46].


**Learning Disability and Other Cognitive Disorders**


According to the literature, even up to 81% of children with neurofibromatosis type 1 (NF1) exhibit moderate to severe impairment in at least one cognitive domain, with nearly 40% of children meeting diagnostic criteria for Attention Deficit Hyperactivity Disorder (ADHD) [42,43,47,48]. The incidence of intellectual disability (full-scale IQ < 70) is slightly elevated compared to the general population, ranging from 4% to 8% [39,40]. Learning disabilities have been found the most common cognitive disorders in NF1, affecting up to 75% of children, particularly in subjects such as mathematics and reading [20,24].


**Other features**


Due to its multisystem involvement—although its presence is not critical for diagnosis– several other clinical symptoms may be observed, such as craniofacial features (macrocephaly and hypertelorism), short stature, hormonal imbalances (e.g., hyperthyroidism or endocrine tumors), vascular abnormalities (e.g., renal artery stenosis, cardiovascular abnormalities, cerebral vasculopathy), and hypertension [24,42,45,49]. A phenotypic overlap with characteristic features of Noonan syndrome is a distinctive NF1 clinical variant known as neurofibromatosis–Noonan syndrome (NFNS) [45,50].

## 2. Materials and Methods


**Clinical examination**


The study was conducted in accordance with the Declaration of Helsinki, with approval from the ethics committee of the Heim Pal National Children’s Institute, Budapest, Hungary (project no. KUT—45/2024)

We retrospectively reviewed clinical, anamnestic, radiological, and genetic data of 204 individuals with at least one criterion for NF1 diagnosis of the National Institutes of Health (NIH), from which 148 patients fulfilled at least two criteria for the diagnosis. All patients were examined at Heim Pal National Pediatric Institute, Budapest, Hungary, between 1 January 2010 and 30 October 2024. A comprehensive dermatological examination was performed, together with accurate neurological, ophthalmologic, orthopedic, and audiological examinations. During the evaluation of the patients, on an individual basis depending on the severity of the disease, symptoms, and age, cerebral/spinal and chest/abdominal MRI, abdominal ultrasonography, skeletal X-ray survey, echocardiograms, and psychological examinations were also undertaken.


**Genetic evaluation**


A total of 204 individuals with suspected or clinically diagnosed NF1 were seen as outpatients and inpatients at the Heim Pal National Pediatric Institute, Budapest, Hungary. A total of 56 patients did not meet the NIH minimum criteria for clinical diagnosis and were therefore not screened (Figure 1). Of the patients with positive clinical diagnoses (148), 76 subjects were not tested or the test results did not arrive until the study was completed because of the limited accessibility of genetic testing facilities in Hungary. A total of 13 patients of the 70 tested subjects were presented with a positive genetic test, but we did not manage to receive permission to use these data. The remaining 57 subjects signed informed consent prior to the performance of genetic testing, which was undertaken in the Department of Medical Genetics, University of Pécs.

Genetic analysis for the study cohort was conducted using two distinct sequencing techniques. For approximately half of the patient samples, Next-Generation Sequencing (NGS) was employed. This high-throughput approach allowed for the comprehensive examination of multiple gene regions simultaneously, providing a broad overview of potential genetic variants.

The remaining patient samples underwent sequencing via the Sanger method, a gold standard for accuracy in validating specific gene mutations. This technique was utilized to confirm variants identified by NGS and to ensure the fidelity of sequencing results in targeted gene regions.

By integrating both NGS and Sanger sequencing methods, we ensured robust and reliable detection of genetic variants, enhancing the overall validity and precision of our findings.

## 3. Results

We retrospectively reviewed data from a cohort of 204 individuals meeting at least one diagnostic criterion for NF1 of the National Institutes of Health (NIH). Among these, 148 subjects fulfilled ≥2 NIH criteria (Figure 1). A genetic examination of 70 individuals was carried out, of whom 58 index cases were confirmed to have an *NF1* pathogenic variant. Out of the 58 positive tests, 13 were not available for inclusion in this study (Figure 1). During the initial genetic test, in 12 patients, a pathogenic variant in the *NF1* gene was not detected (Figure 1). An extended genetic test is planned for these individuals in the future.

The age of enrolled subjects ranged from 1 to 33 years old (mean age is 16 years), with 135 patients (66.2%) falling into the pediatric age group (≤18 years). According to the age distribution of patients, 78 patients (38.2%) were under 12 years of age (49 males, 29 females) (Figure 2d), 57 patients (27.9%) were between 12 and 18 years of age (28 males, 29 females) (Figure 2e), and 69 patients (33.8%) were over 18 years of age (30 males, 39 females) (Figure 2f). Among the 204 patients, 107 (52%) patients were male, and 97 (48%) patients were female (Figure 2a). A positive result of genetic testing was found in a subset of 58 patients. In this group, there were 32 males (55%) and 26 females (45%) (Figure 2b). A total of 45 subjects from these 58 patients with pathogenic variants were available for our research, of whom 21 were male (47%) and 24 were female (53%) (Figure 2c).

Among the 45 patients with detected, available pathogenic variants, causative single nucleotide variants (SNVs) were found in 40 patients (88.89%), and copy number variations (CNVs) in 5 patients (11.1%). In 17 patients (37.7%), small insertions, deletions, or indels resulting frameshifts were identified, in 11 patients (24.4%) nonsense mutations, in 8 patients (17.8%) splice-site mutations, and in 4 patients (8.9%) missense variants (Figure 3).

The analysis revealed some significant associations between specific *NF1 mutations* and various clinical manifestations.


**Skin Manifestations**


Multiple CALMs were observed in all subjects in our cohort (100%), while skinfold freckling was present in 26% (15/58) and 26.7% (12/45) of the patients (Figure 4). Freckling was identified in 25% (7/28) of children under 12 years of age, 15% (3/20) of adolescents aged 12–18 years, and 50% (5/10) of adults over 18 years of age (Table 1). Examining the variant types, the highest prevalence of freckling was among patients with CNVs (80%, 4/5), while subjects with other mutation types did not show a significant correlation (frameshift 3/17, 17.6%, nonsense 3/11, 27.3%, splice-site 2/8, 25%) (Table 2). Patients with missense variants negatively correlated with freckling, although just four patients belonged to this group (Figure 5).


**Neurofibromas and Other Tumors**


Neurofibromas were observed in 16/58 (27.6%) of the patients, of whom seven cases were detected in the youngest generation (25%), eight cases from the age of 12 to 18 years (40%), and one case in the adult group (10%) (Table 1). Neurofibromas were observed in four patients (80%) with CNV, in three patients (37.5%) with splicing, in four patients (36.4%) with nonsense and less frequently, in four patients with frameshift variants (23.5%). No neurofibroma was detected in subjects with missense variants (0/4, 0%) (Table 2).

Plexiform neurofibromas were less common (19%, 11/58, and 17.8%, 8/45), with a notable absence in nonsense (0/11) and missense (0/4) variant carriers. Plexiform neurofibroma was detected in patients with a frameshift mutation (4/17, 23.5%), splice-site mutation (3/8, 37.5%), and CNV (1/5, 20%) (Table 2.). Plexiform neurofibromas were observed in 17.9% (5/28) of patients under 12 years of age, 25% (5/20) of patients aged 12–18 years, and 10% (1/10) of patients over 18 years of age (Table 1).

One patient with microdeletion syndrome developed a malignant peripheral nerve sheath tumor (MPNST), ultimately leading to death. Internal neurofibromas were also seen in this cohort, predominantly in patients with CNV (Table 2).


**Ocular Manifestations**


Lisch nodules were detected in all age groups with a growing incidence: 6/28, 21.4% in patients younger than 12 years, 7/20, 35% in patients between 12 and 18 years, and 6/10, 60% in the adult population (Table 1). Lisch nodules were found in 32.8% (19/58) and 35.6% (16/45) of patients, with splice-site variants demonstrating the highest frequency (50%, 4/8) and missense variants the lowest (0%) (Table 2).

Optic gliomas were reported in 15.5% (9/58) and 13.3% (6/45) of patients. The frequency of optic pathway gliomas was 7.1% (2/28) in the age group under 12 years, 25% (5/20) in the age group between 12 and 18 years, and 20% (2/10) in the age group over 18 years (Table 1). Among the variant types, the highest frequency was in patients with CNV (20%, 1/5), although the results are not significant (Table 2).


**Skeletal Manifestations**


Skeletal abnormalities were found in 27.6% (16/58) and 31.1% (14/45) of patients and were associated with whole gene deletions (3/5, 60%), splicing (4/8, 50%), frameshift (5/17, 29.41%) and nonsense variants (3/11, 27.27%) (Table 2). Scoliosis (13/45, 28.9%) and pectus excavatum (6.6%, 3/45) were notable skeletal manifestations. The age distribution of the research sample was as follows: 14.3% (4/28) of children were under 12 years of age, 45% (9/20) of adolescents aged 12–18 years, and 30% (3/10) of adults were over 18 years of age (Table 1).

Arthralgia or arthritis was not more common in any of the mutation types (whole gene deletions (1/5, 20%), nonsense (2/11, 18.18%), splicing (1/8, 12.5%), and frameshift variants (1/17, 5.8%). Pseudarthrosis was present in 6.9% (4/58) and 6.7% (3/45) of patients, and they were exclusively in splice-site (2/8, 25%) and frameshift variant (1/17, 5.9%) carriers (Table 2). Valgus deformity was present in one patient with a frameshift mutation (1/17, 5.9%).

Patients with the missense variant did not have any skeletal or joint involvement (0/4, 0%).


**Neurological and brain MRI findings**


The presence of specific structural brain lesions found by brain magnetic resonance imaging (MRI) was observed in 34 out of 58 patients (58.6%) and in 25 out of 45 subjects (55.6%). The prevalence of positive cranial MRI was highest in patients with CNV (5/5, 100%), followed by nonsense (7/11, 63.6%), splicing (5/8, 62.5%), missense (2/4, 50%), and frameshift variants (6/17, 35.3%) (Table 2). Notably, intracranial abnormalities were detected across all age groups, with 60.7% (17/28) under 12 years of age, 60% (12/20) of adolescents aged 12–18 years, and 50% (5/10) of adults over 18 years of age at similar percentages (Table 1).

Epilepsy was detected in a small number of patients, affecting 3 out of the subgroup of 58 (5.2%) and 2 out of the subgroup of 45 (4.4%). In the latter subgroup with known mutations, both of the cases of seizures occurred in patients with nonsense variants (2/11, 18.2%) (Table 2). Among the three patients identified from the subgroup of all positive variant findings, one was in the age group of 12–18 (1/20, 5%), while two were in the age group over 18 years (2/10, 20%) (Table 1).


**Learning Disability and Other Cognitive Disorders**


Cognitive/learning disabilities and behavioral disturbances were observed in 22.4% (13/58) and 22.2% (10/45) of patients, while fine motor disturbances and motor development delays were seen in 19% (11/58) and 20% (9/45) of patients. In the former group, there was no significant association with any of the variants (frameshift 4/17, 23.5%; nonsense 3/11, 27.3%; splice-site 1/8, 12.5%; CNV 1/5, 20%, and missense variants 1/4, 25%), while fine motor disturbances and motor development delays were more commonly observed in patients with CNV (60%, 3/5) compared to the other mutation types (frameshift 1/17, 11.8%; nonsense 2/11, 18.2%; splice-site 1/8, 12.5%; and missense variants 1/4, 25%) (Table 2).


**Other Features**


Craniofacial dysmorphism and macrocephaly were found in 8.6% (5/58) and 6.7% (3/45) of patients, with a higher prevalence in patients with CNVs (20%, 1/5) (Table 2).

The endocrinological disorder was detected in only one patient with a frameshift variant (5.9%, 1/17) (Table 2).

Cardiologic involvement was observed in four (6.9%) and three (6.7%) of the patients, of whom two had frameshift mutations (11.8%) and one had nonsense mutation (9.1%) (Table 2).

In first-degree relatives with neurofibromatosis type 1 (NF1), 10 out of 58 patients (17.2%), and 5 out of 45 patients (11.1%), had positive genetic test results concerning the mutations analyzed in the study. The incidence of positive family history was similar among frameshift (3/17, 17.6%), nonsense (1/11, 9.1%), and splicing (1/8, 12.5%) variants; however, patients with CNV and missense mutations did not have any first-degree relatives with NF1 (Table 2).

## 4. Discussion

We reported the results of a 15-year-long study among 204 individuals referred to our Pediatric Hospital with a suspected NF1 clinical diagnosis. A total of 70 patients were tested for variants in the *NF1* gene. We obtained an 82.8% detection rate that is comparable to previous reports [34,46,51,52,53]. From the 58 positive test results, 45 positive genetic test results were available for us to include in our research, which were detected through Sanger Sequencing and Next Generation Sequencing (NGS).

Since our clinic is a children’s hospital, the mean age of patients in our cohort of the 204 patients was 16 years (from 1 to 33 years), of which 135 patients (66.7%) belonged to the pediatric age group (≤18 years).

The nearly equal gender distribution among the genetically tested patients with slight male predominance (55% male, 45% female), which is also observed in the overall cohort (52% male, 48% female), correlates with several studies [45,54]. Although some studies have observed slight female-skewed distributions [42,55], suggesting that the true gender ratio in NF1 populations might vary depending on geographic location or ascertainment methods.

This study examined genotype–phenotype correlations in 45 NF1 patients with confirmed pathogenic variants. The mutational spectrum primarily consisted of single nucleotide variants (SNVs), identified in 40 patients (88.89%), with copy number variations (CNVs) detected in 5 patients (11.11%), reflecting findings in the literature [55,56]. The goal was to explore potential associations between specific mutation categories (frameshift (N = 17, 37.8%), nonsense (N = 11, 24.4%), splice-site (N = 8, 17.8%), missense (N = 4, 8.9%), and CNV (N = 5, 11.1%)), together with the observed frequency of various clinical features within this heterogeneous disorder (Table 2).

**Frameshift variants**, representing the largest group (37.8%) like in most of the studies [7,34,46,54], showed a notable association with several features. The presence of intracranial findings in patients with frameshift mutations was observed in 35.3% (6/17) of the sample, which is lower than the average prevalence of brain abnormalities found in other mutation types (55.5%) and reported in the literature [42,44,46,51]. Lisch nodules were identified in 6 out of 17 patients (35.3%); however, their occurrence did not demonstrate a stronger association with frameshift variants compared to other mutation types reported in former studies [34,35,36]. The presence of scoliosis (29.4%, 5/17) in this group also warrants further investigation given the known association between skeletal abnormalities and NF1 [45]. Although the results obtained in this subgroup statistically were not significant, it is worth noting that pseudarthrosis (5.9%, 1/17) was observed in one patient. Motor developmental delays and fine motor disturbances were observed in 11.8% (2/17) of patients with frameshift variants, while cognitive and learning disabilities were detected in 23.5% (4/17) of patients. These findings are significantly lower than those reported in the literature, similar to our observations in other mutation groups [42,43]. Although cardiovascular malformations are more commonly associated with microdeletion syndrome [24,57], in our cohort, the detection rate of cardiologic involvement (mild mitral valve insufficiency) was the highest in subjects with frameshift mutations at 11.7% (2/17).

The **stop-gain mutation** was the second most common (24.4%) pathogenic mutation type in our cohort, which was consistent with several studies [34], although some other studies reported nonsense mutation to be the most common NF1 pathogenic variant [45,58]. The nonsense mutation group showed a relatively lower frequency of severe manifestations. Intracranial findings were common (63.6%, 7/11), confirming the findings in the literature [42,45,51], but other severe features, such as plexiform neurofibromas, were notably absent. The rates of motor developmental delay/fine motor disturbances (18.2%, 2/11), cognitive/learning disabilities (27.3%, 3/11), and scoliosis (27.3%, 3/11) were similar to those observed in other mutation categories.

The **splice-site mutation** group (17.8%) demonstrated a notable presence of pseudarthrosis (25%, 2/8), suggesting a potential association between this mutation type and this specific skeletal abnormality; however, further investigation in larger cohorts is necessary to establish its clinical significance. Additionally, skeletal abnormalities were observed in 50% (4/8) of patients, alongside a notable presence of intracranial findings in 62.5% (5/8) of patients, aligning with known manifestations of NF1 [7,20,24,51]. Lisch nodules were also reported in 50% (4/8) of patients with splice-site variants, although previous studies have indicated a slightly lower incidence [7,45]. The prevalence of plexiform neurofibromas was the highest in this group (3/8, 37.5%) compared to other mutation types, but the presence of cognitive or learning disabilities was limited to only one case (12.5%), which is remarkably lower than reported in former studies [7]. While we detected notably high percentages of various clinical features within this cohort, the limited sample size necessitates caution in drawing firm conclusions. The greater severity of phenotypes associated with splicing variants compared to frameshift or other loss-of-function variants may be explained by disrupted pre-mRNA processing, leading to aberrant transcripts that evade nonsense-mediated decay (NMD) and exert dominant-negative effects. Factors such as the genetic diversity within our Hungarian cohort, variations in modifier genes or regulatory elements, environmental and epigenetic influences, variant distribution, and methodological differences may also contribute to the observed severity. Notably, similar findings of heightened severity linked to splice-site variants have been reported in Duchenne muscular dystrophy, highlighting the broader significance of this mechanism in genetic disorders [59]. Larger-scale studies are essential to confirm these associations and explore their clinical implications.

In our study, patients with **missense variants** were present with a milder clinical phenotype, which was consistent with the literature, although the small number of patients (n = 4) with missense mutation limits definitive conclusions [15,60,61,62]. While half of patients (50%, 2/4) exhibited NF1-specific intracranial findings, no other striking clinical features like neurofibroma/plexiform neurofibroma (0%, 0/4), skeletal involvement (0%, 0/4), pseudarthrosis (0%, 0/4), optic glioma (0%, 0/4), or Lisch nodule (0%, 0/4) were observed, except the presence of one case of cognitive/learning disabilities (25%, 1/4) and one case of motor developmental delay (25%, 1/4). Larger cohorts will be required to establish further meaningful genotype–phenotype correlations.

**NF1 microdeletion syndrome**, which involves the deletion of large segments of the *NF1* gene and its flanking region, is associated with more severe phenotypes, as evidenced by our findings in a large number of neurofibromas [5,55,56,62,63,64]. Four out of five patients (80%) with CNV had numerous neurofibromas all over the body, including internal ones for instance in the liver and ovarium. In one patient, paravertebral neurofibromas caused aqueductal stenosis leading to hydrocephalus, while another developed a malignant peripheral nerve sheath tumor, resulting in severe neurological symptoms, paresis, and eventually death at the age of 18. Contrary to the literature, plexiform neurofibromas were found in only one (20%) of the patients in our cohort, although whole-body MRI was not routinely performed on all of our patients [31,55,57,62,63,64]. In our study, NF1-specific intracranial findings were detected in all patients (100%), although only one (20%) out of five children with CNV investigated by head MRI had an optic pathway glioma. This finding is consistent with the literature, indicating that there is no increased risk of optic glioma in children with *NF1* microdeletions compared to the general NF1 population [57,63]. As reported in the literature, motor developmental delays and fine motor disturbances were observed in three out of five patients (60%) [57,65,66]. Skeletal abnormalities were detected in three (60%) out of five patients, of whom two (40%) had scoliosis and one (20%) had pectus excavatum. This aligns with previous findings, which reported scoliosis in 43% and pectus excavatum in 30% of patients with microdeletions [67,68]. Valgus deformity was also identified in one (20%) of our patients. Café-au-lait macules were present in all CNV cases, and skinfold freckling also had a high appearance (80%) in this group. Additional clinical manifestations, like dysmorphic facial features, including macrocephaly and Noonan-like facial features, were observed in just one patient (20%) with CNV, which is significantly below the data as reported formerly [5,42,56,57]. In contrast to the literature, cognitive disability was noted in only one patient (20%), which may be attributed to the limited size of our cohort [63].

Our findings demonstrate an age-related increase in the prevalence of **neurofibromas**, and **plexiform neurofibromas** in the age group of 12–18 years compared to the age group under 12 years, which is in line with their natural course in NF1 [20,24,27,28] (Table 1). However, the observed prevalence of these features in the age group over 18 years did not demonstrate a statistically significant increase, possibly due to the limited sample size of adult patients in our primarily pediatric cohort. It is important to note that nearly all of these now-adult patients were children during the data-collection phase of our 15-year study period, which may have a potential bias.

While the prevalence of **Lisch nodules** in our cohort was 32.8% (19/58) and 35.6% (16/45) of patients, consistent with certain studies [45], this is lower than that observed in other reports [20,33,52,63]. This discrepancy might be due to differences in study design, patient selection criteria, or other factors. However, the increasing trend in Lisch nodule prevalence with age observed in this study (6/28, 21.4% in those younger than 12 years, 7/20, 35% in patients between 12 and 18 years, and 6/10, 60% in the adult population) is consistent with the results documented in the literature [20,33]. In contrast to previous findings [34,35,36], our cohort reported the highest frequency of Lisch nodules not in patients with frameshift mutations, but in those with splice-site variants (50%, 4/8). This discrepancy may be attributed to the limited sample size of our cohort.

**Optic gliomas** were detected in 15.5% (9/58) and 13.3% (6/45) of patients, correlating with the literature [20,24,37,38,52]. However, in our cohort, the incidence of optic pathway gliomas was notably higher among patients aged from 12 to 18 years (25%, 5/20) compared to those under 12 years of age (7.1%, 2/28). This finding contradicts the literature suggesting that the majority of optic gliomas are detectable by the age of 7 [20,24,37]. This discrepancy may be attributed to potential documentation deficiencies. Current guidelines recommend annual eye examinations for all patients under 10 years old. Additionally, there is less consensus on screening frequency beyond this age, with some centers recommending evaluations every two years until the age of 18 or continuing for 10 to 25 years following the initial diagnosis [20].

The high prevalence of **NF1-specific brain MR findings** detected in young children (60.7%) and the consistent prevalence across the older age groups (60% and 50% in the 12–18 and >18-year-old groups, respectively) highlight the importance of early neuroimaging in NF1 patients. This suggests that these findings are often present early in the course of the disease, and do not exhibit a substantial age-related increase in prevalence.

In contrast to this, **epilepsy** was detected in the older age group, which can be explained as a consequence of underlying brain abnormalities. The observed cases of seizures, particularly those associated with nonsense variants, highlight the importance of monitoring for neurological complications in this population.

**Skeletal abnormalities** were identified in 27.6% (16/58) and 31.1% (14/45) of patients in our cohort, which is slightly lower than the reported prevalence of 50% [20,24,38,39,40]. However, the detection rates for scoliosis (28.9%, 13/45) and pseudarthrosis (6.7%, 3/45) were consistent with existing literature, indicating a prevalence for scoliosis between 21% and 49%, while pseudarthrosis is reported to occur in approximately 5% of patients [20,24,39,40,69]. Notably, most skeletal abnormalities were observed in the age group of 12–18 years (45%, 9/20), in contrast to a lower incidence in early childhood (14.3%, 4/28), where such abnormalities are typically detected. These findings suggest that, in accordance with the literature, there is a need for more comprehensive orthopedic and radiological examinations in our practice to identify skeletal abnormalities at an early stage.

**Cognitive/learning disabilities and behavioral disturbances** like ADHD (Attention Deficit Hyperactivity Disease) and ASD (Autistic Spectrum Disorder) were observed in 22.4% (13/58) and 22.2% (10/45) in our cohort, which was remarkably lower than that reported in previous studies [42,43,47,48,69]. This difference might be attributed to underdiagnoses, possibly related to the social stigma surrounding ASD and ADHD in Hungary, and limited access to complete patient data due to the encryption of psychological/psychiatric records. Given these challenges, it is essential for all children with NF1 to be closely monitored for developmental delays and behavioral issues.

**Macrocephaly and craniofacial dysmorphism** were reported in only 8.6% (5/58) and 6.7% (3/45) of patients, respectively, which is significantly lower than the 50–75% prevalence rates reported by other authors [42,69,70]. This discrepancy may be attributed to regional variations and potentially inaccurate documentation, emphasizing the need for thorough physical examinations and precise documentation in clinical practice.

**Endocrinological disorders** were significantly underdetected (1/58, 1.7%, and 1/45, 2.2%), suggesting that the actual prevalence may be higher [45]. Therefore, a more frequent and thorough examination in this area is recommended.

Further investigation is required to ascertain whether these observations extend to the other mutational categories identified in this study. The variability in the prevalence of various clinical features across different mutational types underscores the complexities inherent in the NF1 genotype–phenotype relationship. It is important to note that we were unable to adequately assess genotype–phenotype correlations for certain clinically significant complications, such as various cancers, due to their relatively low prevalence in our cohort. This highlights the necessity for larger, more comprehensive studies to fully elucidate the intricate interplay between genotype and phenotype in NF1. Furthermore, these findings pave the way for the future development of personalized follow-up protocols and therapeutic approaches tailored to the specific genotype–phenotype profiles of NF1 patients, potentially improving clinical outcomes.

This study has some limitations that must be considered when interpreting the results. First, the relatively small sample size may reflect both the formerly limited access to genetic testing for NF1 in Hungary, despite recent increases in availability, and the country’s relatively small population (under 10 million), which may have reduced the statistical power to detect subtle genotype–phenotype correlations. This is particularly true for less frequent manifestations, potentially leading to an underestimation of their prevalence. Second, the retrospective nature of this study, encompassing data collected over 15 years, means that patient ages were calculated at the time of manuscript preparation from birth dates, and thus do not reflect the patients’ ages at the time of each clinical examination. This may result in an underreporting of clinical features that could have developed subsequently. Third, the lack of complete clinical assessments, including neurological, ophthalmological, orthopedic, cardiological, and audiological examinations, as cranial and whole-body MRIs for all participants, introduces further limitations that might affect the comprehensive evaluation of genotype–phenotype relationships.

Despite these limitations, this study offers valuable insights into potential correlations between certain NF1 mutations and clinical manifestations. Future research incorporating larger, prospectively collected cohorts with complete clinical and genetic data is necessary to validate these findings and advance our understanding of genotype–phenotype relationships in NF1. One potential solution to address the limitation of sample size is fostering collaborations with other centers to pool data from multiple cohorts. Such multi-center studies would enable a more comprehensive evaluation of NF1 genotype–phenotype correlations, providing a stronger foundation for clinical applications and translational research.

## 5. Conclusions

This study investigated the genotype–phenotype correlations in a cohort of 45 NF1 patients with confirmed pathogenic variants, categorized into frameshift (n = 17), nonsense (n = 11), splice-site variants (n = 8), and CNV (n = 5). While the overall clinical heterogeneity of NF1 is well established, this analysis aimed to identify potential associations between specific mutation types and the frequency of various clinical features.

The genotype–phenotype correlations in NF1 are challenging, due to the extensive clinical variability, the progressive nature of the disorder, and the diverse mutational spectrum. However, these correlations are expected to become increasingly important in managing NF1 patients, particularly when variants are associated with a higher risk of major clinical manifestations and malignancy.

In our cohort, patients with splice-site mutation demonstrated a notably more severe phenotype than in other studies with a high prevalence of plexiform neurofibromas, intracranial findings, skeletal abnormalities, Lisch nodules, and even pseudarthrosis. This suggests a potentially distinct phenotypic presentation of NF1 in the Hungarian population, emphasizing the need for thorough clinical evaluation in patients with splice-site mutations. Further research is needed to investigate population-specific modifying factors.

In alignment with existing literature, our study found that patients with microdeletion syndrome exhibited a more severe clinical phenotype compared to individuals with intragenic NF1 mutations, highlighting the importance of thorough monitoring and follow-up using brain and whole-body MRI to assess for intracranial manifestations, neurofibromas, and potential malignancies.

## Figures and Tables

**Figure 1 biomedicines-13-00146-f001:**
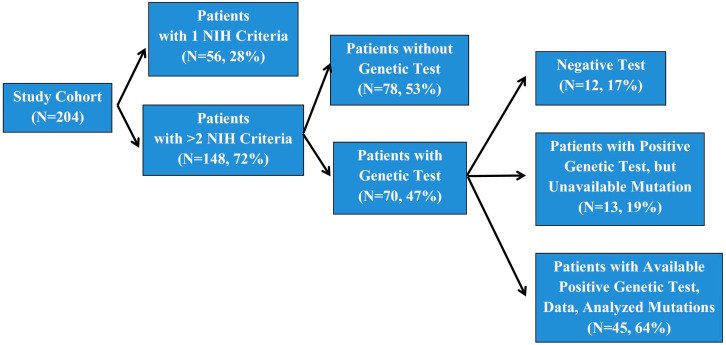
Number of patients involved in our study.

**Figure 2 biomedicines-13-00146-f002:**
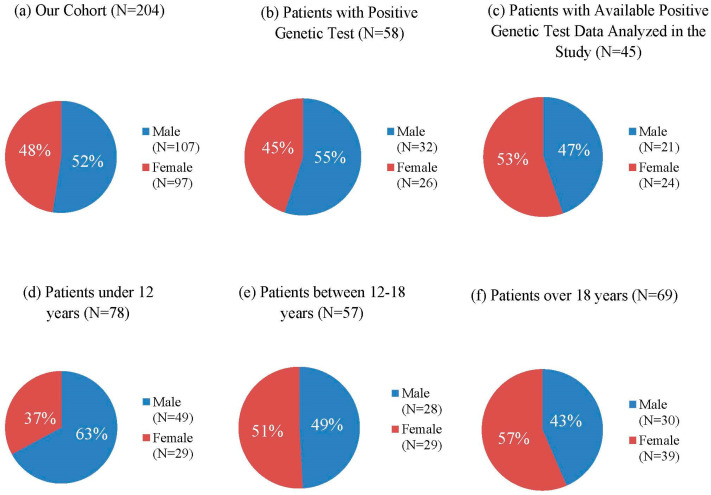
Age and Gender Distribution in NF1 Patients of our Cohort: (**a**) Gender Distribution of our Cohort, (**b**) Gender Distribution of Patients with Positive Genetic Test, (**c**) Gender Distribution of Patients with Available Positive Mutation analyzed in the study, (**d**) Gender Distribution of Patients under 12 years of age, (**e**) Gender Distribution of Patients between 12 and 18 years, (**f**) Gender Distribution of Patients over 18 years of age.

**Figure 3 biomedicines-13-00146-f003:**
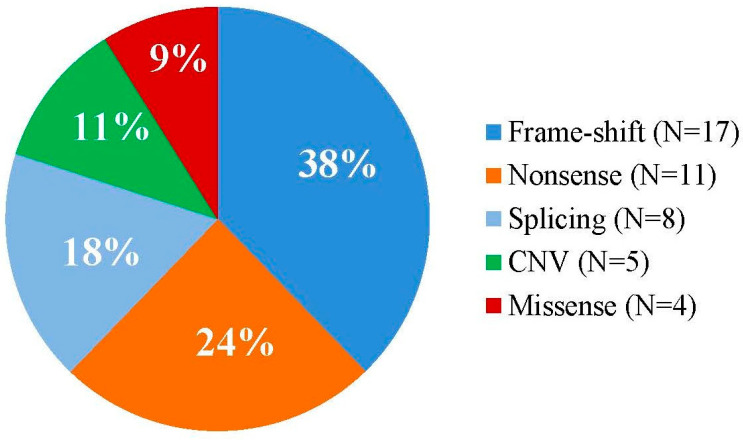
Relative proportion of the different types of variants among the *NF1*-mutations identified.

**Figure 4 biomedicines-13-00146-f004:**
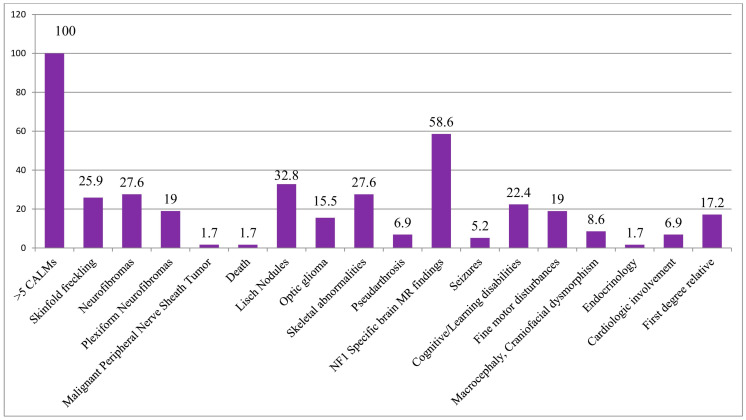
Percentage distribution of the frequency of the most common NF1-related clinical features in the cohort of patients with positive test results.

**Figure 5 biomedicines-13-00146-f005:**
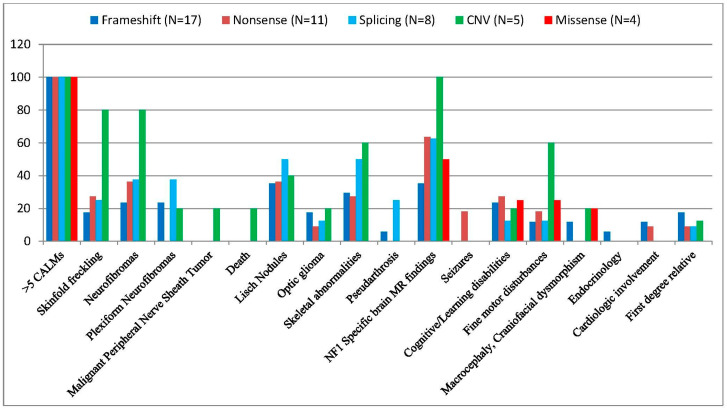
Percentage distribution of the frequency of the most common NF1-related clinical features in the different variant types of our patients. (dark blue color) Frameshift (N = 17), (brown color) Nonsense (N = 11), (light blue color) Splice-Site (N = 8), (green color) CNV (N = 5), (red color) Missense (N = 4).

**Table 1 biomedicines-13-00146-t001:** Prevalence of Clinical Features in NF1 Patients in the following age groups: under 12 years, from 12 to 18 years, and over 18 years.

NF1 Features	All Positive Genetic Test Results (N = 58)	Under 12 Years (N = 28)	12–18 Years (N = 20)	Over 18 Years (N = 10)
**Subjects with positive genetic test**	**58 (100%)**	**28 (100%)**	**20 (100%)**	**10 (100%)**
**Male:female**	**32:26 (55.2%:44.8%)**	**18:10 (64.3%:35.7%)**	**10:10 (50%:50%)**	**4:6 (40%:60%)**
**>5 CALMs**	**58 (100%)**	**28 (100%)**	**20 (100%)**	**10 (100%)**
**Skinfold freckling**	**15 (25.9%)**	**7 (25%)**	**3 (15%)**	**5 (50%)**
**Neurofíbromas**	**16 (27.6%)**	**7 (25%)**	**8 (40%)**	**1 (10%)**
**Plexiform neurofibromas**	**11 (19%)**	**5 (17.9%)**	**5 (25%)**	**1 (10%)**
**Malignant peripheral nerve sheath tumor**	**1 (1.7%)**	**0 (0%)**	**0 (0%)**	**1 (10%)**
**Death**	**1 (1.7%)**	**0 (0%)**	**0 (0%)**	**1 (10%)**
**Lisch nodules**	**19 (32.8%)**	**6 (21.4%)**	**7 (35%)**	**6 (60%)**
**Optic glioma**	**9 (15.5%)**	**2 (7.1%)**	**5 (25%)**	**2 (20%)**
**Skeletal abnormalities**	**16 (27.6%)**	**4 (14.3%)**	**9 (45%)**	**3 (30%)**
**Pseudarthrosis**	**4 (6.9%)**	**1 (3.6%)**	**2 (10%)**	**1 (10%)**
**NF1 specific brain MR findings**	**34 (58.6%)**	**17 (60.7%)**	**12 (60%)**	**5 (50%)**
**Seizures**	**3 (5.2%)**	**0 (0%)**	**1 (5%)**	**2 (20%)**
**Cognitive/learning disabilities, behavioural disturbances**	**13 (22.4%)**	**4 (14.3%)**	**5 (25%)**	**4 (40%)**
**Fine motor disturbances, motor developmental delay**	**11 (19%)**	**5 (17.9%)**	**5 (25%)**	**1 (10%)**
**Macrocephaly, craniofacial dysmorphism**	**5 (8.6%)**	**4 (14.3%)**	**0 (0%)**	**1 (10%)**
**Endocrinology**	**1 (1.7%)**	**0 (0%)**	**0 (0%)**	**1 (10%)**
**Cardiologic Involvement**	**4 (6.9%)**	**2 (7.1%)**	**2 (10%)**	**0 (0%)**
**First degree relative**	**6 (10.3%)**	**3 (10.7%)**	**3 (15%)**	**0 (0%)**

**Table 2 biomedicines-13-00146-t002:** Prevalence of Clinical Features in Neurofibromatosis Type 1 Patients Classified by Frameshift, Nonsense, Splice-Site, CNV, and Missense Mutations (Percentages calculated within each variant subgroup).

NF 1 Features	Mutation-Positive Individuals (N = 58)	Available, Analysed Mutations (N = 45)	Frameshift (N = 17)	Nonsense (N = 11)	Splicing (N = 8)	CNV (N = 5)	Missense (N = 4)
**Male:female**	**32:26** **(55.2%:44.8%)**	**21:24** **(46.7%:53.3%)**	**8:7** **(53.3%:6.7%)**	**4:7** **(36.4%:63.6%)**	**4:4 (50%:50%)**	**1:4 (20%:80%)**	**3:1** **(75%:25%)**
**>5 CALMs**	**58 (100%)**	**45 (100%)**	**17 (100%)**	**11 (100%)**	**8 (100%)**	**5 (100%)**	**4 (100%)**
**Skinfold freckling**	**15 (25.9%)**	**12 (26.7%)**	**3 (17.6%)**	**3 (27.3%)**	**2 (25%)**	**4 (80%)**	**0 (0%)**
**Neurofibromas**	**16 (27.6%)**	**15 (33.3%)**	**4 (23.5%)**	**4 (36.4%)**	**3 (37.5%)**	**4 (80%)**	**0 (0%)**
**Plexiform neurofibromas**	**11 (19%)**	**8 (17.8%)**	**4 (23.5%)**	**0 (0%)**	**3 (37.5%)**	**1 (20%)**	**0 (0%)**
**Malignant peripheral nerve sheath tumor**	**1 (1.7%)**	**1 (2.2%)**	**0 (0%)**	**0 (0%)**	**0 (0%)**	**1 (20%)**	**0 (0%)**
**Death**	**1 (1.7%)**	**1 (2.2%)**	**0 (0%)**	**0 (0%)**	**0 (0%)**	**1 (20%)**	**0 (0%)**
**Lisch** **nodules**	**19 (32.8%)**	**16 (35.6%)**	**6 (35.3%)**	**4 (36.4%)**	**4 (50%)**	**2 (40%)**	**0 (0%)**
**Optic glioma**	**9 (15.5%)**	**6 (13.3%)**	**3 (17.6%)**	**1 (9.1%)**	**1 (12.5%)**	**1 (20%)**	**0 (0%)**
**Skeletal abnormalities**	**16 (27.6%)**	**14 (31.1%)**	**5 (29.4%)**	**3 (27.3%)**	**4 (50%)**	**3 (60%)**	**0 (0%)**
**Pseudarthrosis**	**4 (6.9%)**	**3 (6.7%)**	**1 (5.9%)**	**0 (0%)**	**2 (25%)**	**0 (0%)**	**0 (0%)**
**NF1 specific brain MR findings**	**34 (58.6%)**	**25 (55.6%)**	**6 (35.3%)**	**7 (63.6%)**	**5 (62.5%)**	**5 (100%)**	**2 (50%)**
**Seizures**	**3 (5.2%)**	**2 (4.4%)**	**0 (0%)**	**2 (18.2%)**	**0 (0%)**	**0 (0%)**	**0 (0%)**
**Cognitive/learning disabilities**	**13 (22.4%)**	**10 (22.2%)**	**4 (23.5%)**	**3 (27.3%)**	**1 (12.5%)**	**1 (20%)**	**1 (25%)**
**Fine motor disturbances**	**11 (19%)**	**9 (20%)**	**2 (11.8%)**	**2 (18.2%)**	**1 (12.5%)**	**3 (60%)**	**1 (25%)**
**Macrocephaly, craniofacial dysmorphism**	**5 (8.6%)**	**3 (6.7%)**	**2 (11.8%)**	**0 (0%)**	**0 (0%)**	**1 (20%)**	**0 (0%)**
**Endocrinology**	**1 (1.7%)**	**1 (2.2%)**	**1 (5.9%)**	**0 (0%)**	**0 (0%)**	**0 (0%)**	**0 (0%)**
**Cardiologic involvement**	**4 (6.9%)**	**3 (6.7%)**	**2 (11.8%)**	**1 (9.1%)**	**0 (0%)**	**0 (0%)**	**0 (0%)**
**First degree relative**	**10 (17.2%)**	**5 (11.1%)**	**3 (17.6%)**	**1 (9.1%)**	**1 (12.5%)**	**0 (0%)**	**0 (0%)**

## Data Availability

The data presented in this study are available on request from the corresponding author.

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
