# Peer review of "Increased Phenotype Severity Associated with Splice-Site Variants in a Hungarian Pediatric Neurofibromatosis 1 Cohort: A Retrospective Study"

_biomedicines, 2025, doi:10.3390/biomedicines13010146_

Round 1

Reviewer 1 Report

Comments and Suggestions for Authors

The authors report the results of a very interesting retrospective study they conducted in a cohort of 204 Hungarian individuals with NF1, aiming at identifying potential genotype-phenotype correlations. Their data showed that splicing variants in the NF1 gene are associated with severe phenotypes, even more severe than reported in previous studies. The article is interesting and add important information to the current knowledge about the elusive genotype-phenotype correlations in a complex disorder such as NF1.

I would suggest some revisions to further improve the overall clarity:

1) I suggest the manuscript is revised by a native English speaker, as there are some language inaccuracies throughout the text. This should be fixed;

2) Please avoid the term ‘mutation’ and use ‘variant’ in its place, throughout the manuscript;

3) It would be beneficial for the readers if the authors could increase the dimension of the characters employed in the Figures and Tables, which now look hardly readable;

4) How the authors would explain the high severity of phenotypes associated with splicing variants compared to other loss-of-function variants, such as frameshift? 

5) Also, the presented data suggest that NF1 phenotypes associated with splicing defects in this cohort are more sever compared to previous studies. Can the authors provide some hypotheses in this regard, discussing what could explain this finding?

Comments on the Quality of English Language

The language could be improved throughout the manuscript with little effort.

Author Response

Dear Reviewer 1,

We thank you for your thorough review of our manuscript and for your positive feedback regarding our study. We appreciate your valuable suggestions, which have helped us improve the quality and clarity of our manuscript. Below, we address each of your comments in detail:

  1. I suggest the manuscript is revised by a native English speaker, as there are some language inaccuracies throughout the text. This should be fixed;

We agree with your suggestion to revise the manuscript for language accuracy. The manuscript has been thoroughly reviewed by two English language lecturers, and all language inaccuracies have been corrected to ensure clarity and readability. Corrections are highlighted in red letters.

2) Please avoid the term ‘mutation’ and use ‘variant’ in its place, throughout the manuscript;

We acknowledge your comment regarding the use of the term "mutation" and have replaced it in most of the cases with "variant" throughout the manuscript (highlighted in red letters). While "mutation" is often used in the literature in similar contexts, we understand the importance of consistent and modern terminology, and we appreciate your recommendation.

3) It would be beneficial for the readers if the authors could increase the dimension of the characters employed in the Figures and Tables, which now look hardly readable;

The font size in all figures and tables has been increased, as suggested. We agree that this adjustment significantly improves their readability and presentation. We thank you for highlighting this issue.

4) How the authors would explain the high severity of phenotypes associated with splicing variants compared to other loss-of-function variants, such as frameshift?

The greater severity of phenotypes associated with splicing variants compared to frameshift or other loss-of-function variants can be attributed to the mechanisms involved. Splicing variants often disrupt the proper processing of pre-mRNA, leading to aberrant transcripts that may evade nonsense-mediated decay (NMD). These aberrant transcripts can result in dominant-negative effects or other deleterious interactions that exacerbate phenotypic severity. In contrast, frameshift variants generally lead to truncated proteins that are more likely to be degraded by NMD, thereby minimizing their pathological impact. This mechanistic difference may explain the heightened severity observed in cases with splicing variants. (highlighted in purple letters).

5) Also, the presented data suggest that NF1 phenotypes associated with splicing defects in this cohort are more sever compared to previous studies. Can the authors provide some hypotheses in this regard, discussing what could explain this finding?

We agree that the phenotypes associated with splicing defects in our cohort appear to be more severe compared to previously reported studies. This discrepancy could be due to several factors:

  • Genetic Background: The genetic diversity within our Hungarian cohort may influence the phenotypic outcomes of NF1 splicing variants. Variations in modifier genes or regulatory elements could exacerbate the impact of splicing defects.
  • Environmental and Epigenetic Factors: Differences in environmental exposures or epigenetic modifications could contribute to the more severe manifestations observed in our cohort.
  • Variant Distribution: The specific splicing variants identified in our study may differ from those in other cohorts, potentially affecting the observed phenotypic spectrum.
  • Methodological Differences: The use of more sensitive or comprehensive diagnostic techniques in our study may have captured more subtle or severe phenotypes that were underreported in prior research.

We have included these hypotheses in the discussion section of the revised manuscript and referenced relevant studies to support our explanations. (highlighted in purple letters).

Comments on the Quality of English Language
We have ensured that the language has been reviewed and improved throughout the manuscript, as suggested, and we are confident that it now meets a high standard of clarity.

Thank you once again for your constructive feedback. We are confident that the revisions have enhanced the quality of our manuscript and addressed all of your concerns.

Sincerely,
Klara Veres

Reviewer 2 Report

Comments and Suggestions for Authors  

Here’s a simplified and humanized review paragraph that incorporates constructive feedback for the authors:

This study provides valuable insights into genotype-phenotype correlations in NF1 patients, utilizing both NGS and Sanger sequencing to ensure reliable genetic analysis. However, certain aspects could benefit from improvement. The clinical relevance of the findings, particularly how they can influence patient care or treatment, should be more clearly emphasized. While the use of a long-term dataset is a strength, the inconsistent use of diagnostic tools, such as MRI scans, limits the ability to fully understand the clinical manifestations. Furthermore, data presentation could be improved by incorporating more visual aids, such as graphs or tables, to make the results clearer and easier to interpret. The discussion does a good job addressing limitations but could expand on potential solutions, such as increasing sample size through collaboration with other centers. Overall, this study is a significant contribution, but addressing these points would enhance its clarity, impact, and applicability.

Comments on the Quality of English Language

The English language is clear but could benefit from minor grammatical corrections and simplification for improved readability

Author Response

Dear Reviewer 2,

We sincerely thank you for taking the time to thoroughly review our manuscript and provide thoughtful and constructive feedback. Your positive comments and suggestions are highly appreciated, as they have greatly contributed to improving the quality of our work. Below, we address your points in detail:

  1. The clinical relevance of the findings, particularly how they can influence patient care or treatment, should be more clearly emphasized.
    The clinical relevance of our findings lies in their potential to improve patient care and treatment by highlighting the importance of incorporating genotype-phenotype correlations into clinical practice. In the revised manuscript, we have also emphasized the future possibilities of personalized follow-up and therapeutic approaches informed by these correlations. We believe this clarification strengthens the practical applicability of our study. (highlighted in blue letters).
  2. While the use of a long-term dataset is a strength, the inconsistent use of diagnostic tools, such as MRI scans, limits the ability to fully understand the clinical manifestations.

We acknowledge your concern regarding the inconsistent use of diagnostic tools, particularly MRI scans. While MRI examinations are indeed critical for detailed clinical evaluations, they were not available for all patients in our cohort due to retrospective data collection and resource limitations. We have now discussed this limitation in greater depth in the revised manuscript and proposed strategies for addressing it in future studies.

3.Furthermore, data presentation could be improved by incorporating more visual aids, such as graphs or tables, to make the results clearer and easier to interpret.

In response to your suggestion, we have improved the visual presentation of our data. Specifically, we have increased the font sizes across all figures to enhance readability. Additionally, we combined Figure 5 into a more comprehensive format, which now provides a clearer and more comparable view of the individual variations. These changes significantly improve the accessibility and interpretability of our results.

  1. The discussion does a good job addressing limitations but could expand on potential solutions, such as increasing sample size through collaboration with other centers.

We fully agree that increasing the sample size would strengthen the study's findings and their generalizability. While this was not feasible in our current study due to logistical constraints, we acknowledge the importance of multi-center collaborations for pooling larger datasets. This approach is highlighted as a key recommendation for future research in the revised discussion section. (highlighted in green letters).

Comments on the Quality of English Language
We have carefully revised the manuscript to address minor grammatical issues and simplify the language where appropriate, ensuring improved readability and clarity.

Thank you once again for your valuable insights and constructive feedback. We are confident that these revisions have significantly improved the clarity, impact, and applicability of our manuscript.

Sincerely,
Klara Veres

Round 2

Reviewer 2 Report

Comments and Suggestions for Authors

Dear Authers,

I would like to confirm that all the reviewer’s comments for the manuscript titled "Increased Phenotype Severity Associated with Splice-Site Mutations in a Hungarian Pediatric Neurofibromatosis 1 Cohort: A Retrospective Study" (Manuscript ID: biomedicines-3409273) have been addressed thoroughly and satisfactorily.

Thank you for your time and guidance.

Best regards,

Comments on the Quality of English Language

The manuscript has been carefully revised for grammar, spelling, and clarity to ensure high linguistic quality